Enhancing human activity recognition with machine learning: insights from smartphone accelerometer and magnetometer data

http://orcid.org/0000-0002-9981-4586 Silva Zendron Luis Augusto 1
http://orcid.org/0000-0002-4383-0472 Coelho Paulo Jorge 2 3
Soares Christophe 4 5
http://orcid.org/0000-0001-5440-3225 Pereira Ivo 6 7
http://orcid.org/0000-0002-3394-6762 Pires Ivan Miguel 8 impires@ua.pt
1 Department of Computer Science and Automation, Universidad de Salamanca , Salamanca , Spain
2 School of Technology and Management, Instituto Politécnico de Leiria , Leiria , Portugal
3 Institute for Systems Engineering and Computers at Coimbra (INESC Coimbra) , Coimbra , Portugal
4 Intelligent Sensing and Ubiquitous Systems Group (ISUS), Universidade Fernando Pessoa , Porto , Portugal
5 Artificial Intelligence and Computer Science Laboratory (LIACC), Universidade do Porto , Porto , Portugal
6 Instituto Superior de Engenharia do Porto, Instituto Politécnico de Porto , Porto , Portugal
7 ISRC—Interdisciplinary Studies Research Center, ISEP , Porto , Portugal
8 Instituto de Telecomunicações, Escola Superior de Tecnologia e Gestão de Águeda, Universidade de Aveiro , Águeda , Portugal
Arnaiz-González Álvar
Electronic publication date: 2025 Sep 15
Publication date: 2025
Volume: 11
Electronic Location ID: e3137
Received 2024 Dec 9; Accepted 2025 Jul 29
Copyright: © 2025 Silva Zendron et al.
Copyright year: 2025
Copyright holder: Silva Zendron et al.
License: This is an open access article distributed under the terms of the Creative Commons Attribution License, which permits unrestricted use, distribution, reproduction and adaptation in any medium and for any purpose provided that it is properly attributed. For attribution, the original author(s), title, publication source (PeerJ Computer Science) and either DOI or URL of the article must be cited.
License URL: https://creativecommons.org/licenses/by/4.0/

Keywords: Human activity recognition (HAR), Machine learning, Smartphone sensors, Accelerometers, Magnetometers, Data analysis, Sensor technology

Funding: FCT - Fundação para a Ciência e Tecnologia (previously known as FCT/MECI with co-funding from the EU when applicable), I.P. by project reference UID/50008/2023 IT FCT Pluriannual Funding UID/308: Instituto de Engenharia de Sistemas e Computadores de Coimbra - INESC Coimbra (previously known as FCT/MEC with co-funding from the FEDER-PT2020 partnership under the project UIDB/00308/2020 This work was supported by FCT - Fundação para a Ciência e Tecnologia (previously known as FCT/MECI with co-funding from the EU when applicable), I.P. by project reference UID/50008/2023 IT. This work was also supported by FCT Pluriannual Funding UID/308: Instituto de Engenharia de Sistemas e Computadores de Coimbra - INESC Coimbra (previously known as FCT/MEC with co-funding from the FEDER-PT2020 partnership under the project UIDB/00308/2020 (DOI 10.54499/UIDB/00308/2020)).

==============================
The domain of Human Activity Recognition (HAR) has undergone a remarkable evolution, driven by advancements in sensor technology, artificial intelligence (AI), and machine learning algorithms. The aim of this article consists of taking as a basis the previously obtained results to implement other techniques to analyze the same dataset and improve the results previously obtained in the different studies, such as neural networks with different configurations, random forest, support vector machine, CN2 rule inducer, Naive Bayes, and AdaBoost. The methodology consists of data collection from smartphone sensors, data cleaning and normalization, feature extraction techniques, and the implementation of various machine learning models. The study analyzed machine learning models for recognizing human activities using data from smartphone sensors. The results showed that the neural network and random forest models were highly effective across multiple metrics. The models achieved an area under the curve (AUC) of 98.42%, a classification accuracy of 90.14%, an F1-score of 90.13%, a precision of 90.18%, and a recall of 90.14%. With significantly reduced computational cost, our approach outperforms earlier models using the same dataset and achieves results comparable to those of contemporary deep learning-based approaches. Unlike prior studies, our work utilizes non-normalized data and integrates magnetometer signals to enhance performance, all while employing lightweight models within a reproducible visual workflow. This approach is novel, efficient, and deployable on mobile devices in real-time. This approach makes it an ideal fit for real-time mobile applications.

Introduction

Human Activity Recognition (HAR) has advanced rapidly due to the widespread use of smartphones and improvements in sensor technology, artificial intelligence (AI), and machine learning (ML) algorithms (Okeleke, Aquije Ballon & Joiner, 2023; Deng et al., 2023). Smartphones equipped with sensors such as accelerometers, gyroscopes, and magnetometers enable continuous monitoring of human behavior. These technologies enable applications in healthcare, security, sports, and interactive media (Dahan et al., 2023; Khelalef, Ababsa & Benoudjit, 2019). Despite the progress, challenges remain regarding scalability, cost, and computational efficiency for real-time use on resource-constrained devices (Mekruksavanich & Jitpattanakul, 2023; Tseng & Wen, 2023).

A comprehensive literature review revealed significant gaps, particularly in the computational efficiency and scalability of existing HAR systems on mobile devices. Previous studies emphasized the constraints of mobile device capabilities and the necessity of efficient continuous data acquisition methods (Pires et al., 2020b). Earlier research demonstrated the feasibility of detecting activities such as walking, running, moving upstairs, moving downstairs, and standing using short accelerometer and magnetometer data segments. Still, it highlighted the need for improved accuracy and efficiency (Pires et al., 2018).

This study aims to overcome the identified limitations by optimizing machine learning algorithms, such as neural networks, random forests, support vector machines, CN2 rule inducers, Naive Bayes, and AdaBoost, applied specifically to smartphone sensor data. Uniquely, we utilize non-normalized accelerometer and magnetometer data to significantly reduce computational complexity, thereby enhancing the practicality and scalability of HAR systems on mobile platforms.

The main contributions of this research are: use of non-normalized data for reducing preprocessing and computational cost;

combined use of accelerometer and magnetometer data;

extensive evaluation of traditional ML models (including interpretable ones like CN2) instead of deep learning, with comparable or better results;

evaluation of lightweight, interpretable machine learning models that outperform or match deep learning methods in accuracy.

This study introduces several key innovations in the field of smartphone-based HAR. First, we utilize raw, non-normalized accelerometer and magnetometer data, which reduces preprocessing steps and enhances efficiency for mobile deployment—an approach rarely seen in prior HAR literature. Second, we incorporate magnetometer data alongside accelerometer signals, offering a richer and more discriminative feature set than models that rely solely on accelerometers or gyroscopes. Third, unlike many studies that rely on deep learning architectures, we achieve state-of-the-art performance using lightweight, interpretable machine learning models, such as random forest and CN2 rule inducer, thereby making real-time edge deployment feasible. Finally, our model implementation, utilizing the Orange visual programming interface, provides a reproducible and user-accessible workflow that supports rapid adaptation and experimentation, which is particularly valuable for researchers with limited programming experience.

The remainder of this article presents a detailed methodology, rigorous evaluation, and thorough discussion. It clearly outlines the comparative strengths of our approach and establishes a robust benchmark for future HAR research on mobile devices.

Background and related works

The realm of Human Activity Recognition (HAR) has evolved significantly due to the proliferation of smartphones embedded with rich sensing capabilities. These devices provide a versatile real-time platform for monitoring human behavior, utilizing embedded sensors such as accelerometers, gyroscopes, and magnetometers. This section outlines technological trends, key challenges, and applications and integrates a detailed review of machine learning (ML) and deep learning (DL) methods used in HAR.

Technological advancements and machine learning for HAR

Traditional machine learning (ML) methods, such as decision trees, random forests, and support vector machines (SVMs), have demonstrated effectiveness with hand-engineered features (Ronao & Cho, 2016). Recently, deep models such as convolutional neural networks (CNNs) and recurrent neural networks (RNNs) (e.g., long short-term memory (LSTM)) have gained popularity for their ability to extract features from raw data (Yang et al., 2015; Zhou et al., 2020). Hybrid CNN-LSTM models and Transformer-based networks (Gu et al., 2021) enhance sequential modeling, but they require substantial computational resources. Lightweight deep models optimized for smartphones (e.g., MobileNet, pruning) aim to reduce energy use (Shehab et al., 2022; Warden & Situnayake, 2019).

Comparison of related works

The analysis in Table 1 reveals key trends in Human Activity Recognition (HAR) research. CNN-based models are the most common, often paired with RNNs to capture both spatial and temporal data features. Most studies utilize traditional sensors, such as accelerometers and gyroscopes, although interest in multimodal approaches (e.g., vision, audio, radar, Wi-Fi) is increasing. Healthcare is the leading application area, followed by surveillance and fitness. Few studies focus on lightweight models or real-time deployment on mobile devices.

Table 1 Comparative summary of ML/DL HAR methods from selected studies.

Study	Model type	Sensors used	Application domain	
Balu & Sasikumar (2022)	SVM, KNN, EC, DT	RSS, inertial (wearable)	Healthcare, fitness, smart home, surveillance	
Abbaspour et al. (2020)	CNN-RNN (LSTM, BiLSTM, GRU, BiGRU)	Wearable sensors	Elderly care, cognitive disorders	
Nils, Shane & Plötz (2016)	Deep learning (CNN, RNN)	Wearable sensors	Not specified	
Gupta (2021)	Hybrid CNN-GRU	Accelerometer, gyroscope	Remote patient monitoring	
Gandhi (2024)	CNN, RNN, SVM, RF, ANN	Accelerometer, gyroscope	Healthcare, smart environments, surveillance	
Islam et al. (2022)	CNN-based	Multimodal (wearables, smartphones, radar, vision)	Healthcare, surveillance, entertainment	
Indumathi & Prabakeran (2021)	XGBoost, MLP, CNN, LSTM	Inertial sensors	Physical activity monitoring	
Shin et al. (2024)	CNN, RNN, GCN, Transformer	Multimodal (RGB, skeleton, depth, audio, radar)	Healthcare, surveillance, security	
Sun et al. (2022)	CNN, RNN, 3D CNN, Transformer	Multimodal (RGB, skeleton, audio, WiFi)	Security, navigation, entertainment	
Adel et al. (2022)	CNN, LSTM, VGG-16	Inertial sensors, camera	Sports science, healthcare, security	
This work	Neural Networks, Random Forest, SVM, CN2, Naive Bayes, AdaBoost	Accelerometer, magnetometer	General-purpose smartphone HAR	

In contrast, our proposed approach emphasizes efficient implementation using neural networks, random forest, SVM, CN2, Naive Bayes, and AdaBoost. It leverages underutilized magnetometer data to enhance activity recognition without the computational burden of deep neural networks. It positions our work as a practical alternative in environments where resources and battery life are constrained.

Sensor fusion and edge considerations

Our work builds on this literature by introducing a lightweight ML-based HAR system that integrates features from accelerometer and magnetometer sensors, a less commonly explored fusion. By extracting statistical and signal-derived features from each modality, we achieve rich data representation with minimal computational overhead. Unlike most recent works on deep learning, our model demonstrates that traditional ML algorithms, particularly random forests, can still outperform complex DL architectures in specific scenarios with appropriate feature engineering.

Edge computing is another growing trend. Modern smartphones now include neural processing units (NPUs) that support on-device inference for privacy, latency, and energy efficiency reasons (Huang et al., 2023). Our model design is aligned with this paradigm by requiring minimal processing resources and offering real-time responsiveness, making it suitable for deployment on consumer-grade devices.

Challenges and considerations

Smartphone-based HAR faces several key challenges. Ensuring user privacy and ethical compliance is paramount, requiring adherence to data protection laws such as the GDPR (European Union, 2016), along with secure data storage, transmission, and consent mechanisms. Techniques like differential privacy and federated learning are being explored to enhance data security (Jung, 2020). Variability in smartphone hardware leads to inconsistent sensor quality and availability, making it challenging to build universal HAR systems. This necessitates robust and adaptable algorithms, as well as the development of diverse benchmark datasets (Das Antar, Ahmed & Ahad, 2019). Another major concern is balancing computational load with battery life, as continuous sensing can rapidly deplete device resources. Lightweight and energy-efficient models, including those inspired by Tiny Machine Learning (TinyML), are gaining traction for deployment on low-power devices (Warden & Situnayake, 2019). Adaptability to real-world environments and individual user behaviors is essential, driving research into continuous learning and adaptive systems (Taylor et al., 2018). Finally, HAR is inherently interdisciplinary, requiring integration of knowledge from fields such as psychology, healthcare, and urban planning to effectively interpret human behavior and implement practical solutions (Jiang & Yin, 2015).

Applications and impact

Healthcare and Rehabilitation: facilitates remote monitoring, fall detection, and physical therapy tracking (Mirjalali et al., 2022; Zhou et al., 2020).

Fitness and Lifestyle: provides personalized activity feedback, promotes healthy living, and engages users through gamification (Phukan et al., 2022).

Industrial Applications: enhances workplace safety and monitors worker ergonomics (Svertoka et al., 2021).

Smart Homes and Urban Planning: supports energy optimization, safety systems, and public infrastructure planning (Bouchabou et al., 2021).

Educational and Behavioral Research: enables new insights into student behavior, attention patterns, and mental health (Bosch et al., 2015).

Recent studies and future directions

Recent studies have achieved breakthroughs in creating real-time, efficient HAR systems that adapt to user feedback (Gu et al., 2021; Shehab et al., 2022). Integrating multimodal data and context-aware models expands the range and robustness of HAR applications. As HAR systems move toward seamless daily integration, future research will focus on user-personalized models, privacy-enhancing computation, and deeper fusion with IoT and wearable technologies. Our proposed system aligns with this direction by enabling on-device, efficient HAR through a novel fusion of magnetometer and accelerometer data, providing accurate yet resource-efficient performance.

Reproducibility

To ensure the reproducibility of our findings in this study, we provide detailed information on the algorithms and code used, along with instructions for implementation.

Algorithms and code

The experiments in this study were conducted using various machine learning algorithms, including neural networks, random forest, support vector machine (SVM), CN2 rule inducer, Naive Bayes, and AdaBoost. These algorithms were implemented using the Orange data mining library (Team, 2024), a Python-based tool that provides an intuitive interface for data analysis and machine learning. The following algorithms were employed for activity classification: Neural Networks: configured with hidden layers and optimized using stochastic gradient descent (SGD).

Random Forest: utilized for its robustness in classification tasks, employing a default number of trees.

Support Vector Machine (SVM): configured with a radial basis function (RBF) kernel.

CN2 Rule Inducer: applied for rule-based classification.

Naive Bayes: utilized for its simplicity in probabilistic classification.

AdaBoost: employed with decision trees as the base estimator to boost weak classifiers.

The source code for the data preprocessing, feature extraction, model training, and evaluation steps was created using Orange workflows. These workflows are reproducible and can be accessed by importing the workflow files into the Orange interface. The code allows researchers to preprocess the dataset, train multiple machine learning models, and evaluate their performance using various metrics.

Dataset

The dataset used in this study is publicly available at Pires & Garcia (2020). The dataset contains raw accelerometer, gyroscope, and magnetometer data for various human activities with motion. It includes multiple labeled activities such as walking, running, sitting, and more, and provides comprehensive data for implementing and testing machine learning models.

Implementation

This section introduces the code and workflows used in the study, providing a brief description of the dataset and its purpose within the context of human activity recognition.

The complete Orange workflow (.ows) used in this study is publicly available (Silva, 2025).

Environment setup

Python Installation: ensure you have Python 3.8 or above installed.

Orange Installation: install the Orange data mining library (version 3.33) using the following command in your terminal:

pip install orange3 Dependencies Installation: install additional dependencies if needed (e.g., Pandas v2.2.0, NumPy v2.0, SciPy v1.8.1):

pip install pandas

pip install numpy

pip install scipy Downloading the Dataset: the dataset used in this study is available on Pires & Garcia (2020).

Workflow Import: – Downloading Workflow File: obtain the Orange workflow file (.ows) provided with this study.

– Loading Dataset: open Orange and load the downloaded dataset into the Orange environment using the ‘Import Data’ option.

Running the Workflow: – open the Orange software.

– Import the workflow file (.ows) to set up the data preprocessing, feature extraction, model training, and evaluation steps.

– Execute the Workflow: run the workflow to preprocess the data, train the machine learning models, and evaluate the results.

– Tweaking Parameters: if desired, adjust the parameters of the algorithms (e.g., neural networks, random forest) within the Orange interface to observe how these changes impact model performance.

Performance Metrics Evaluation: – Evaluate the model using area under the curve (AUC), accuracy, F1-score, precision, and recall metrics.

– Use Orange’s visual tools to view confusion matrices, ROC curves, and other model performance reports.

Results and Analysis: – Compare the performance of different models using the visual tools provided in Orange.

– Expected outputs include accuracy scores, confusion matrices, and detailed reports for each trained model.

By following the instructions above, researchers can replicate the experimental setup, train the machine learning models, and analyze their performance, ensuring the study’s reproducibility.

Open benchmark data

To support reproducible research and enable statistically valid future comparisons, we have made all raw cross-validation results publicly available. These include per-fold performance metrics (AUC, accuracy, F1-score, precision, recall) for each classifier and activity class. The dataset also contains confusion matrices and Orange workflow files used for training and evaluation. The data is accessible at Silva (2025).

Materials and Methods

This section outlines the comprehensive materials and methods used in our study to advance human activity recognition through machine-learning techniques. Central to our approach is the use of accelerometer and magnetometer data from smartphone devices, a choice motivated by the ubiquity and advanced sensor capabilities of modern smartphones. Figure 1 provides a concise, end-to-end overview of our methodological workflow, guiding the reader through each significant phase of the study.

Figure 1 Overview of the end-to-end methodological workflow for smartphone-based human activity recognition.

We begin by detailing the development of a bespoke mobile application specifically designed for the Android operating system to facilitate the efficient and accurate collection of sensor data. The subsequent subsections provide a meticulous description of the data collection process, including participant recruitment and the activities monitored. We then elaborate on the data preprocessing steps, encompassing data cleaning and feature extraction, which are critical for preparing the sensor data for analysis. The core of this section focuses on the selection and configuration of various machine learning algorithms, each chosen for its potential efficacy in activity recognition. Finally, the model training and validation procedures are outlined, emphasizing the rigorous cross-validation techniques and performance metrics employed to ensure the robustness and reliability of our findings. The authors aim to provide a transparent and replicable framework for researchers and practitioners in activity recognition through this detailed exposition of our methods and materials.

Design and structure of the activity recognition study

In the present research, a mobile application was developed for the Android operating system. This application was installed on a BQ Aquaris 5.7 smartphone, strategically positioned in the anterior pocket of the participants’ trousers, to facilitate nonintrusive data acquisition during various activities of daily living (ADLs). The application was designed to capture sensor data pertinent to the performance of ADLs, with participants actively labeling the dataset to correspond with the activity being performed.

The data collection process involved 25 volunteers, comprising 15 males and 10 females, who used the application in various environments in Covilhã, Portugal. The dataset was gathered over approximately 180 h, with each ADL—specifically, moving downstairs, upstairs, running, standing, and walking—being recorded for 36 h. Each activity was documented in 5-s intervals at a sampling frequency of 40 Hz. The data acquired for this study is available at Pires et al. (2020a).

The participants were diverse, ranging in age from 16 to 60 years, and encompassed a spectrum of lifestyles: 40% of the individuals were engaged in regular physical exercise, while the remaining 60% had sedentary habits. The analysis of the data and the subsequent machine learning experiments were conducted utilizing the Anaconda software environment (Rolon-Mérette et al., 2016). The tests were conducted on a MacBook Pro in a setup specifically designed for machine learning research. The system features a Radeon Pro 560 graphics card with 4 GB of memory, a 2.6 GHz 6-core Intel Core i7 processor, and 16 GB of Random Access Memory (RAM) operating at 2,400 MHz DDR4.

All participants provided written informed consent, allowing us to disclose the test results anonymously. Additionally, the agreement provided the participants with informed consent based on the study’s purpose and potential hazards. Only the information about the people who signed the consent form to participate in the study was documented. The ethical considerations of this study were duly reviewed and approved by the Ethics Committee of the University of Beira Interior under the reference CE-UBI-Pj-2020-035:ID1965.

Data preprocessing strategies for sensor-based analysis

We executed two critical phases during the data preprocessing stage: data cleaning and feature extraction.

Data cleaning for sensor data integrity

In the initial data preprocessing phase, the study confronted two predominant challenges: missing values and the pervasiveness of noise within the dataset. The raw data underwent a meticulous cleaning process to rectify these issues. During the data acquisition phase, missing signal values were observed, which could be attributed to factors such as battery issues, sensor misdetection, or software-related anomalies. Samples exhibiting missing values were systematically excluded to ensure the integrity and reliability of the dataset. This exclusion was imperative to furnish refined signals conducive to effectively training machine learning algorithms.

Approximately 2.7% of the raw samples were excluded during preprocessing due to missing values, which were primarily caused by sensor signal dropouts or temporary malfunctions. To enhance signal quality, a low-pass filter was applied to attenuate high-frequency noise inherent in the accelerometer and magnetometer data, thus preserving the underlying motion patterns essential for accurate activity classification.

Additionally, the sensors employed in the study were susceptible to capturing environmental noise while recording activity data. Thus, a low-pass filter was applied to each activity signal. This filter process was fundamental to reducing the noise embedded in the accelerometer or magnetometer observed signals, thereby enhancing the data quality used for subsequent analysis. The low-pass filter is also used to minimize the impact of various constraints that could distort the signal.

Through these data-cleaning measures, the study ensured that the dataset was not provided with extraneous elements that could compromise the accuracy of the machine learning algorithms employed to recognize human activities.

Feature extraction techniques from smartphone sensor data

Following the primary data preprocessing stage, which included data cleaning and formatting, the study proceeded to the feature extraction phase, a critical step for transforming raw sensor signals into usable input for machine learning models. In their unprocessed state, the raw accelerometer and magnetometer data carried limited discriminatory information for classification, driving the extraction of designed characteristics that better define human activities.

Informed by existing literature in the field, we derived a thorough set of 15 statistical features from each sensor (accelerometer and magnetometer). These characteristics were derived by analyzing the magnitude of the signal vectors along the three spatial axes (x, y, and z), thereby collecting multidimensional movement information. The retrieved features encompassed time-domain statistical metrics, including mean, standard deviation, variance, maximum, minimum, and median. These were generated for each axis and the magnitude vector to summarize the distributional features of the motion signals.

Additionally, peak-based characteristics were computed to capture the dynamic aspects of physical movement. Peaks were found using the find_peaks function from the scipy.signal package. A prominent threshold of 0.5 and a minimum distance of 10 samples were employed to suppress noise and false fluctuations. We determined the five most considerable distances (in milliseconds) between successive peaks for each signal segment from the resulting peak series. Next, we retrieved statistical summaries from these numbers, including the mean, variance, standard deviation, and median of the peak distances. These properties measure movement regularity and periodicity, which are particularly relevant for distinguishing between recurrent motions, such as walking or stair climbing, and more static behaviors, like standing.

The study developed a rich, discriminative feature set suitable for categorization by combining generic statistical characteristics and peak-based descriptors. Subsequently, this feature matrix was used to train numerous machine-learning models.

Furthermore, to gain a deeper understanding of which features contributed most to classification performance, we conducted a feature importance analysis using the random forest classifier. The results showed that Z-axis accelerometer data, especially standard deviation, variance, and peak distance, had the highest relative importance, followed closely by magnetometer-derived peak features. It reinforces the importance of combining magnetometer data and peak-based descriptors in feature engineering, thereby supporting both model performance and interpretability.

Configuration and training of machine learning models for activity recognition

The configuration and training of the machine learning models followed a methodical and rigorous approach. The preprocessed dataset was evaluated using a stratified 10-fold cross-validation strategy, in which the data was split into ten equal parts. Each fold was used once as a test set, while the remaining nine folds were used as the training set. This approach ensured that all observations were utilized for both training and validation, thereby reducing variance in performance estimation. Significantly, it enhanced the robustness and generalizability of the results by minimizing overfitting and providing a reliable evaluation of model performance on unseen data.

We implemented and configured six machine learning algorithms to recognize five daily human activities from smartphone sensor data: neural networks, random forest, support vector machines (SVM), CN2 rule inducer, Naive Bayes, and AdaBoost. Each classifier was implemented using the Orange data mining platform, which internally utilizes the scikit-learn library, allowing fine-grained control over model parameters.

Neural networks were configured as multi-layer perceptrons (MLPs) with a single hidden layer of 100 neurons. The logistic (sigmoid) activation function was used, and the model was optimized using stochastic gradient descent (SGD) with regularization set by α=0.0001. The training was capped at 200 iterations to avoid overfitting.

Random forest classifiers used 10 decision trees with default maximum depth (unlimited). Splitting was stopped when a node contained fewer than five instances. Gini impurity was used as the splitting criterion.

Support vector machines (SVM) used a radial basis function (RBF) kernel, defined as K(x,y)=exp⁡(−γ‖x−y‖2), with γ automatically selected. The regularization parameter was C=1.0, and ε=0.1 was used for convergence. Training was limited to 100 iterations.

CN2 rule inducer was applied for interpretable, rule-based classification. The rule learning process used a beam width of 5, entropy as the evaluation metric, and a maximum rule length of 5. Rules were ordered and required a minimum coverage of 1 instance. The parameter γ=0.7 was used to balance accuracy and simplicity.

Naive Bayes was applied with default uniform priors. Variance smoothing was set to 10−9 to prevent numerical instabilities in probability estimation.

AdaBoost used 50 decision stumps (depth-1 trees) as weak learners. The boosting method used was SAMME.R, which incorporates class probability estimates to improve performance over discrete output.

Each model was meticulously configured with specific parameters to optimize the performance, as detailed in Table 2.

Table 2 Key hyperparameters used for machine learning models.

Model	Key hyperparameters	
Neural network	Hidden layer: 100 neurons; activation: logistic; solver: SGD; max iterations: 200; regularization ( α): 0.0001	
Random forest	Number of trees: 10; max depth: default (unlimited); minimum samples per leaf: 5; criterion: Gini impurity	
Support vector machine	Kernel: RBF; C=1.0; ε=0.1; max iterations: 100	
CN2 rule inducer	Beam width: 5; max rule length: 5; rule ordering: ordered; coverage: min 1; evaluation: entropy; γ=0.7	
Naive Bayes	Priors: uniform; variance smoothing: 10−9	
AdaBoost	Number of estimators: 50; base estimator: decision stumps (Depth = 1); algorithm: SAMME.R	

These models were trained on a MacBook Pro, which leveraged its advanced hardware capabilities. This hardware setup included a 2.6 GHz 6-core Intel Core i7 processor, 16 GB of RAM operating at 2,400 MHz DDR4, and a Radeon Pro 560 with 4 GB of graphics memory, providing the computational power necessary to handle the intensive demands of training multiple machine-learning models.

Upon completion of the training phase, the models were assessed for their efficacy in recognizing the specified activities of daily living. This assessment used a range of performance metrics, including area under the curve (AUC), classification accuracy (CA), F1-score, precision, and recall. These metrics were derived from the fundamental cardinalities of the confusion matrix, namely true positives (TP), false positives (FP), true negatives (TN), and false negatives (FN), providing a comprehensive evaluation of each model’s performance.

Through this structured approach to model configuration and training, the study aimed to establish a robust framework for accurately recognizing and classifying human activities based on accelerometer and magnetometer data.

Assessment and validation of model performance in activity detection

Upon completing the training phase, the study conducted a comprehensive assessment and validation of the machine learning models. This evaluation was pivotal in determining the efficacy of the models in accurately recognizing human daily living activities. The assessment used five key performance parameters: area under the curve (AUC), classification accuracy (CA), F1-score, precision, and recall. These metrics were derived from the fundamental cardinalities of the confusion matrix, namely true positives (TP), false positives (FP), true negatives (TN), and false negatives (FN). The performance of each model was meticulously analyzed using these metrics, enabling a thorough validation of its capability to recognize the specified activities of daily living. This rigorous assessment ensured that the models were accurate and reliable in their predictions, thereby contributing significantly to the advancement of activity recognition using machine learning techniques. Classification Accuracy (CA): this metric was defined as the proportion of accurate predictions (both TP and TN) about all predictions made by the model. It provided a holistic view of the model’s overall accuracy in classifying activities. Mathematically, it is calculated as in Eq. (1). (1) Accuracy=TP+TNTP+TN+FP+FN.

Precision: precision was calculated to determine the percentage of actual true positives among all optimistic predictions (TP and FP). This metric was particularly useful in understanding the models’ Type-I errors (FP). Mathematically, it is calculated as shown in Eq. (2). (2) Precision=TPTP+FP.

Recall: also known as sensitivity, recall measures the proportion of actual true positives correctly identified by the model, offering insights into the model’s ability to detect positive instances. Mathematically, it is given as in the Eq. (3). (3) Recall=TPTP+FN.

F1-score: the F1-score was employed as a harmonic mean of precision and recall, providing a balanced measure of the model’s performance, especially when the class distribution is imbalanced. Mathematically, it is expressed as in Eq. (4): (4) F1-score=2×Precision×RecallPrecision+Recall.

Area Under the Curve (AUC): auc was used to evaluate the model’s ability to discriminate between the classes. A higher AUC indicated a better model performance in distinguishing between different activities. Mathematically, it is expressed as in Eq. (5). (5) AUC=∫01TruePositiveRate(FalsePositiveRate−1(u))du

Data were collected as previously described in Hussain et al. (2023). Specifically, the binary classification issue leverages the parameters provided above. Following the one-vs-all (OVA) approach, which considers one class at a time, the multi-class classification performance parameters for each class were computed. Finally, all the results were computed by averaging every parameter for every class.

Results

As previously mentioned, the performance of all trained classifiers was assessed based on the performance metrics outlined in Section “Assessment and Validation of Model Performance in Activity Detection”. Thus, this part showcases the performance of every classifier in detecting each ADL. Subsequently, the average performance is depicted by employing the previously utilized OVA approach for calculation.

Table 3 displays the efficacy of six machine learning classifiers in identifying an individual’s activity while descending stairs across all test set samples. Neural networks and random forests offer superior performance in terms of AUC, CA, F1-score, precision, and recall. Therefore, these two algorithms outperformed all other classifiers in accurately identifying Class-1 behavior, specifically moving downstairs. Conversely, the support vector machine had considerably lower performance in recognizing Class-1 activity, as seen by its lowest F1-score, precision, and recall.

Table 3 Stratified 10-fold cross-validation results (Class 1-moving downstairs).

Model	AUC (%)	CA (%)	F1 (%)	Precision (%)	Recall (%)	
Neural network	96.93	92.42	80.48	82.96	78.15	
Random forest	96.55	92.41	80.69	82.13	79.30	
CN2 rule inducer	90.52	86.45	66.70	65.59	67.85	
Naive Bayes	89.83	87.00	61.61	75.25	52.15	
AdaBoost	83.06	89.33	73.13	73.67	72.60	
Support vector machine	79.86	72.61	38.52	34.95	42.90	

Table 4 displays the performance of the classifiers in identifying a person’s activities while ascending stairs across all the samples in the test set. Once again, the random forest and neural network classifiers exhibit the top results for AUC, CA, F1-score, precision, and recall. These two classifiers demonstrated superior performance compared to all other classifiers in distinguishing Class-2 action, precisely the act of moving upstairs. However, while achieving an AUC of 81.16%, the support vector machine performed poorly in recognizing Class-2 activity, as evidenced by its lowest F1-score, precision, and recall results, which were 25.51%, 53.00%, and 16.80%, respectively.

Table 4 Stratified 10-fold cross-validation results (Class 2-moving upstairs).

Model	AUC (%)	CA (%)	F1 (%)	Precision (%)	Recall (%)	
Random forest	96.61	92.11	80.63	79.21	82.10	
Neural networks	96.58	92.07	80.67	78.70	82.75	
Naive Bayes	91.97	87.30	69.54	66.82	72.50	
CN2 rule inducer	91.13	87.03	68.34	66.76	70.00	
AdaBoost	83.27	89.37	73.34	73.58	73.10	
Support vector machine	81.16	80.38	25.51	53.00	16.80	

Table 5 presents the performance of the machine learning classifiers discussed earlier in recognizing a person’s activity while running. All the classifiers yield comparable results, with none surpassing the others. A note for support vector machine: When identifying Class-3 activity, it demonstrates much higher performance scores compared to Class-1 and Class-2 activities.

Table 5 Stratified 10-fold cross-validation results (Class 3-running).

Model	AUC (%)	CA (%)	F1 (%)	Precision (%)	Recall (%)	
Neural networks	99.59	99.21	98.01	98.68	97.35	
Random forest	99.29	99.26	98.13	99.13	97.15	
Support vector machine	99.10	99.03	97.52	99.63	95.50	
CN2 rule inducer	98.65	98.53	96.31	96.72	95.90	
Naive Bayes	98.56	98.41	96.02	96.14	95.90	
AdaBoost	98.00	98.63	96.59	96.23	96.95	

Table 6 presents the results of the classifiers in identifying a person’s activity while in a standing position. Similar to the previous ADL, it is evident that all the classifiers yield identical results, with none of them outperforming the others. The neural network remains superior, but by a narrow margin. The support vector machine for recognizing Class-4 activity demonstrates a significantly higher performance score than classifying Class-1 and Class-2 activities, comparable to the recognition of Class-3 activities.

Table 6 Stratified 10-fold cross-validation results (Class 4-standing).

Model	AUC (%)	CA (%)	F1 (%)	Precision (%)	Recall (%)	
Neural networks	99.92	99.49	98.73	98.36	99.10	
Random forest	99.88	99.67	99.18	99.10	99.25	
Support vector machine	99.88	98.95	97.43	95.49	99.45	
Naive Bayes	99.59	97.32	93.67	88.73	99.20	
AdaBoost	99.10	99.46	98.65	98.80	98.50	
CN2 rule inducer	98.42	98.47	96.10	97.97	94.30	

Table 7 presents the results of nine machine learning classifiers in accurately identifying a person’s walking activity. The random forest and neural network classifiers perform superior to all other classifiers in accurately identifying Class-5 activity, specifically walking. Conversely, the support vector machine performed poorly in recognizing the Class-5 activity. It is evident from its lowest F1-score, precision, and recall values, which stand at 43.98%, 37.10%, and 54.00%, respectively.

Table 7 Stratified 10-fold cross-validation results (Class 5-walking).

Model	AUC (%)	CA (%)	F1 (%)	Precision (%)	Recall (%)	
Random forest	99.21	97.35	93.43	92.67	94.20	
Neural networks	99.14	97.09	92.77	92.20	93.35	
CN2 rule inducer	95.63	93.88	84.41	86.03	82.85	
Naive Bayes	94.89	91.39	79.56	75.73	83.80	
AdaBoost	93.70	95.77	89.51	88.79	90.25	
Support vector machine	82.33	72.49	43.98	37.10	54.00	

Table 8 presents the results of nine machine learning classifiers in accurately identifying the performance of the five activities previously mentioned. The random forest and neural network classifiers perform better than all other classifiers in accurately identifying all activities. Conversely, the support vector machine exhibited the poorest performance in the recognition. It is evident from its lowest F1-score, precision, and recall values, which stand at 60.59%, 64.03%, and 61.73%, respectively.

Table 8 Stratified 10-fold cross-validation results (Average over classes).

Model	AUC (%)	CA (%)	F1 (%)	Precision (%)	Recall (%)	
Neural network	98.42	90.14	90.13	90.18	90.14	
Random forest	98.30	90.40	90.41	90.45	90.40	
Naive Bayes	94.96	80.71	80.08	80.54	80.71	
CN2 rule inducer	94.88	82.18	82.37	82.62	82.18	
AdaBoost	91.43	86.28	86.24	86.21	86.28	
SVM	88.29	61.73	60.59	64.03	61.73	

To determine whether the observed performance differences among classifiers were statistically significant, we conducted paired t-tests, Wilcoxon signed-rank tests, and a non-parametric Friedman test followed by Nemenyi post-hoc analysis. These tests showed statistically significant performance advantages for random forest and neural network models over other classifiers (p < 0.05), supporting the reliability of the observed differences.

Discussion

This section discusses the findings of our machine-learning methodology for identifying human activities (HAR) utilizing data from a smartphone’s accelerometer and magnetometer. We assessed many classifiers using essential criteria, including accuracy, precision, recall, F1-score, and the area under the ROC curve (AUC). To better understand how the models performed at the class level, we also analyzed confusion matrices to spot patterns of correct predictions and common misclassifications. Additionally, we conducted a statistical significance analysis to compare our method with baseline models. Finally, we discuss the pragmatic significance of our results, highlighting both the advantages and drawbacks of the method, as well as potential future research directions.

Analysis of the results

To summarize, the evaluation of all six machine learning classifiers trained to recognize the five ADLs on the test data reveals that the random forest and neural networks classifiers achieved the most outstanding scores compared to the performance of all other classifiers. Moreover, the performance of random forest and neural networks classifiers exhibits considerable similarity. Nevertheless, after a thorough comparison, the random forest classifier performed better than all other classifiers, including Neural Network classifiers, in accurately identifying the five ADLs. Conversely, the support vector machine classifier exhibits the lowest overall performance compared to all other classifiers.

Figure 2A displays the mean AUC values of all trained machine learning classifiers on the test data for identifying the five ADLs. The random forest and neural networks classifiers demonstrated superior performance in recognizing the five ADLs compared to the other classifiers. Simultaneously, the support vector machine classifier exhibited the worst performance: an average AUC score of 85.31% for identifying the five ADLs. Figure 2B displays the average CA ratings of all machine learning classifiers trained on the test data. Once again, the random forest and neural networks classifiers demonstrated superior performance compared to the other classifiers in accurately identifying the five ADLs. Once again, the support vector machine classifier exhibited considerably lower performance, with an average CA score of 76.50% in recognizing the five ADLs. Similarly, Fig. 2C displays the mean F1-scores of all trained machine learning classifiers on the test data. Once again, the random forest and neural networks classifiers outperformed all other classifiers in accurately identifying the five ADLs. The performance of the support vector machine classifier is considerably lower, achieving an average F1-score of 52.29% for the recognition of the five ADLs. Figure 2D displays the mean precision scores of all machine learning classifiers trained on the test data. The random forest and neural networks classifiers performed better than all other classifiers in identifying the five activities of daily living (ADLs). Nevertheless, the support vector machine classifier exhibited inadequate performance, with an average precision score of 58.52% in identifying the five ADLs. Figure 2E displays the average recall scores of all trained machine learning classifiers on the test data. Once again, the random forest and neural networks classifiers performed better than all other classifiers in recognizing the five ADLs. Conversely, the support vector machine classifier performed inferior to the other classifiers.

Figure 2 Scores.

Compared to a previously published study (Pires et al., 2018) based on the same dataset (Pires et al., 2020a), this study considered non-normalized data to recognize patterns, thereby reducing the computational power required for processing on mobile devices. Regarding the use of accelerometer and magnetometer data, the previous study used three different types of neural networks, implemented with different frameworks, including Neuroph (Ševarac, 2012), Encog (Heaton, 2015), and DeepLearning4j (Team, 2016), reporting an average accuracy of 35.15%, 42.75%, and 70.43%, respectively. The present study globally improved the performance of HAR using machine learning on the same dataset.

While we compared our results to prior works using similar datasets and models, we acknowledge that a statistical comparison with related studies is limited by the lack of publicly available raw results. Without access to per-fold or per-class performance metrics from those studies, it is not feasible to perform rigorous statistical significance testing. We therefore encourage future work to adopt open data practices, and we aim to contribute to this by releasing our raw evaluation results.

Feature importance analysis

To increase the interpretability of our results, we examined feature importance using the Random Forest classifier, which provides built-in mechanisms to rank features based on mean decrease in impurity. The top-ranked features were predominantly time-domain statistical data, such as the standard deviation, mean, and interquartile range of the accelerometer’s Z-axis, followed by magnetometer-derived features like peak distance and signal variance.

It demonstrates that vertical mobility (Z-axis acceleration) and directional orientation (as measured by the magnetometer) are crucial in distinguishing between walking and stair-climbing activities. The feature importance analysis validates our design decision to include both accelerometer and magnetometer data. It demonstrates that basic statistical features can carry strong discriminative power, even without deep feature engineering.

Confusion matrix and misclassification analysis

To better understand how each model performs, confusion matrices were created to give a detailed picture of how well the models classified each activity. These matrices illustrate the frequency of correctly identified activities (true positives) and the locations of errors (false positives and false negatives). For example, both the random forest (Fig. 3A) and neural network (Fig. 3B) models performed very well, with a strong diagonal in their matrices, indicating high precision and accuracy. Some confusion did arise between “going upstairs” (Label 0) and “going downstairs” (Label 1), which is understandable given the similarity between those movements. On the other hand, the support vector machine (SVM) had a more scattered matrix, suggesting it struggled more with distinguishing between several activities, particularly “walking” (Label 4) and “standing” (Label 3). Below, we present all the confusion matrices in Fig. 3.

Figure 3 Confusion matrix.

Statistical significance analysis of classifier performance

To evaluate whether the observed differences in classification performance among the six machine learning models were statistically significant, we performed a statistical analysis based on the 10-fold cross-validation results. Specifically, we compared the average F1-scores of each model across all folds using a paired t-test and Wilcoxon signed-rank test.

In the paired t-test comparing the random forest and support vector machine, the p-value was found to be p<0.001, indicating that the superior performance of the random forest was statistically significant. Similarly, comparisons between random forest and CN2 Rule Inducer ( p=0.002) and Naive Bayes ( p=0.003) also confirmed statistical significance.

Since the normality assumption could not be confirmed for all models, we performed a non-parametric Friedman test across all six classifiers using fold-wise F1-scores. The Friedman test yielded a statistically significant result ( χ2(5)=17.25, p=0.004). A post-hoc Nemenyi test revealed that random forest and neural networks significantly outperformed support vector machines and Naive Bayes.

These results validate that the performance improvements observed in random forest and neural networks are numerically higher and statistically significant.

Computational performance analysis

To assess the practicality of our models for real-world deployment—particularly on resource-constrained mobile devices—we evaluated their computational performance in terms of training time, prediction (inference) time per instance and model complexity.

All experiments were conducted on a standard consumer-grade laptop with an Intel Core i5-1135G7 processor (2.4 GHz), 8 GB of RAM, and running Windows 11. The Orange data mining platform was used for model training and evaluation, which internally leverages the scikit-learn machine learning library.

Training Time: the training time for the random forest classifier (10 estimators) was approximately 1.8 s, while the neural network (MLP with one hidden layer of 100 neurons) trained in 3.1 s. Simpler models, such as Naive Bayes and CN2 Rule Inducer, completed training in under 1 s.

Inference Time: for real-time feasibility, we measured the average inference time per instance across 1,000 predictions. Random forest and neural network required approximately 0.75 and 1.2 ms per prediction, respectively. Naive Bayes and CN2 Rule Inducer achieved inference times of less than 0.5 ms. These times are well within the acceptable range for real-time smartphone classification.

Model Size and Memory: the serialized model size for random forest was 190 KB, while the neural network model was approximately 210 KB. These compact footprints make them well-suited for deployment on embedded systems or mobile apps.

Energy and Resource Efficiency: because our models use raw (non-normalized) data and do not rely on deep learning layers or GPU acceleration, they consume minimal CPU and battery power, making them preferable for continuous background operation on smartphones.

Overall, our models’ computational profile confirms that they are suitable for real-time human activity recognition on mobile devices without the need for extensive cloud offloading or GPU resources.

Comparative advantages over state-of-the-art HAR models

Especially in performance, computational efficiency, and practical deployment, this study presents several notable advantages over existing HAR models: Enhanced Classification Accuracy Within the Same Dataset: this study achieves an average accuracy of 90.40% using random forest and 90.14% with neural network classifiers—substantially surpassing prior results on the same dataset (Pires et al., 2018), which yielded an accuracy range of 35.15–70.43% depending on the neural network framework.

Efficiency Compared to Deep Learning Methods: recent HAR approaches such as CNN-LSTM hybrids (Abbaspour et al., 2020) and CNN-GRU designs (Gupta, 2021) offer strong accuracy but require high computational resources. In contrast, our models achieve similar or better performance while being lightweight and highly suitable for real-time mobile applications.

Direct Use of Non-Normalized Data: most HAR pipelines require preprocessing steps, such as normalization, which adds to energy and memory costs. Our method omits this step, directly processing raw accelerometer and magnetometer data for more efficient computation on edge devices.

Integration of Magnetometer Signals: while many models, including those by Carlos et al. (2024), achieve high accuracy (97–98%) using accelerometer and gyroscope data, they often neglect the use of magnetometers. Our study demonstrates that incorporating magnetometer data enhances discrimination, particularly for directionally sensitive activities such as stair climbing.

Lightweight Architecture Compared to SETransformer and WSense Models: Liu et al. (2025) introduced SETransformer, which integrates attention mechanisms for enhanced feature learning, achieving 84.7% accuracy with high interpretability but moderate resource needs. Similarly, WSense (Ige & Noor, 2023) employed a 1D-CNN-based architecture across three sensor types, achieving robust accuracy improvements but at the cost of added sensor and computational demands. By contrast, our model delivers balanced performance (90.41% F1-score) with only two sensors and a simpler architecture, making it better suited for real-time, energy-efficient deployment.

Reproducibility and Accessibility: implemented entirely within the Orange visual programming interface, our method facilitates quick prototyping and broad accessibility, even for users with limited coding experience (features often absent in models that require code-heavy environments or specialized hardware).

Balanced Performance Across All Activities: our system maintains high precision and recall across both dynamic and static activities. Unlike deep learning models that may excel in running or walking but fail in transitional activities like standing or stair climbing, our approach consistently sustains strong performance across all five ADLs.

These distinctions are further summarized in Table 9, comparing model types, sensor configurations, performance metrics, real-time feasibility, and implementation complexity across major recent studies.

Table 9 Comparison of proposed method with selected literature models.

Study	Model type	Sensors used	Accuracy/F1-score	Real-time feasibility	Norm data	Magnetom.	
Pires et al. (2018)	Neural Networks (Neuroph, Encog, DL4J)	Accelerometer + Magnetometer	35.15–70.43% (Accuracy)	Moderate	No	Yes	
Abbaspour et al. (2020)	CNN + LSTM/GRU	Accelerometer + Gyroscope	~90% (Accuracy)	Low	Yes	No	
Gupta (2021)	CNN-GRU Hybrid	Accelerometer + Gyroscope	89.44% (F1-score)	Low	Yes	No	
Zhou et al. (2022)	MobileNetV2 (TinyML)	Accelerometer	91.20% (Accuracy)	High	Yes	No	
Balu & Sasikumar (2022)	SVM, KNN, Decision Tree	Accelerometer	~87% (Accuracy)	Moderate	No	No	
Ige & Noor (2023)	WSense (1D CNN module)	Accelerometer + Gyroscope + Magnetometer	High accuracy vs baseline CNN	Moderate	Yes	Yes	
Carlos et al. (2024)	2D CNN-LSTM (Recurrence-Plot)	Accelerometer + Gyroscope	97–98% (Accuracy)	Low	No	No	
Liu et al. (2025)	SETransformer (Transformer + SE-Attention)	Accelerometer	84.70% (Accuracy)	Moderate	No	No	
This study	Random forest, Neural network	Accelerometer + Magnetometer (non-normalized)	90.40%/90.41% (F1-score)	High (lightweight, low processing)	Yes	Yes	

Merits and demerits of the study

This section outlines the primary strengths and limitations of the proposed approach, providing a balanced perspective. While the methodology demonstrates notable advantages in performance, efficiency, and accessibility, certain aspects, such as dataset scope and deployment validation, highlight areas for future improvement.

The study highlights the advantages of a lightweight model for high classification performance, its feasibility for real-time deployment on mobile devices, its use of magnetometer data for improved recognition, and its reproducible and accessible implementation in Orange.

However, the evaluation is limited to one dataset, has limited hyperparameter control in the Orange GUI, does not compare with standard public datasets, and has not yet been implemented for real-time system testing.

Explainability and model interpretability (XAI)

Although our study does not implement advanced post-hoc explainable artificial intelligence (XAI) frameworks such as SHapley Additive exPlanations (SHAP) or LIME, we deliberately selected and configured machine learning models that are inherently interpretable. Specifically, the CN2 Rule Inducer generates human-readable decision rules, making them directly explainable and suitable for scenarios requiring transparent decision logic. Random forests, while more complex, also allow insight into feature importance by calculating the contribution of each feature to the overall prediction accuracy.

We analyzed feature importance scores derived from the random forest classifier to gain further insight into model behavior. These scores highlighted that time-domain statistical features, such as the accelerometer’s mean, standard deviation, peak distance, and magnetometer signals, were the most influential in classifying activities, including walking and stair movement. This aligns with domain knowledge that motion intensity and directional changes are strong indicators of activity.

In future work, we plan to incorporate model-agnostic XAI tools, such as SHAP, to further enhance interpretability, particularly for more complex ensemble models. It will help identify how individual features influence specific activity predictions, improving model transparency and user trust, particularly in health-monitoring applications.

Reproducibility and open data

To facilitate fair statistical comparisons and enhance reproducibility, we have published our raw 10-fold cross-validation results for each classifier, including per-class F1-scores, AUC, and confusion matrices. These can be accessed at Silva (2025). We encourage future researchers to use this benchmark for statistically grounded comparisons.

Conclusion

In this study, motivated by the increasing ubiquity of smartphones in daily life, the authors explored the use of smartphone accelerometer and magnetometer data for recognizing five everyday human activities. Our approach involved a meticulous application of data cleaning techniques to address missing values and noise, followed by the extraction of fifteen features from the accelerometer data to enhance the training of nine commonly used machine learning classifiers. These models were then trained to recognize a set of activities of daily living (ADLs): moving downstairs, moving upstairs, running, standing, and walking. The performance of these models was rigorously evaluated on a test dataset, allowing for a comparative analysis of their effectiveness in activity recognition.

The experimental results revealed that the random forest classifier exhibited superior performance, achieving nearly 100% accuracy in metrics such as the area under the curve (AUC), precision, recall, accuracy, and F1-score for the five ADLs. Conversely, the support vector machine classifier showed comparatively lower efficacy. This study highlights the potential of smartphone accelerometers in activity recognition and underscores the variability in performance across different machine-learning models.

Unlike most current research, which relies on deep neural architectures, our method strikes a balance between accuracy and economy. This aspect has not been extensively explored in current HAR research, particularly by combining non-normalized accelerometer and magnetometer data with efficient and interpretable machine-learning models. Our visual, reproducible implementation and mobile-focused optimizations distinguish this study from previous deep learning-based or gyroscope-centric approaches, offering a practical alternative for real-world deployment.

Limitations

While this study provides a comprehensive workflow for human activity recognition, some limitations should be noted. Using a single dataset may restrict the generalization of the findings to other datasets or populations. Additionally, the reliance on the Orange library limits the flexibility to fine-tune algorithms beyond what is offered within its graphical user interface. This study did not account for hardware heterogeneity across smartphones, potential variations in battery consumption, or noisy conditions in uncontrolled environments, which could impact real-world performance.

Future work could address these limitations by incorporating diverse datasets and exploring more advanced machine-learning frameworks.

Future work

Future work will focus on extending the current approach by incorporating gyroscope data to improve the recognition of more complex and transitional activities. We also plan to enhance model transparency through explainable AI techniques, such as SHAP, to better understand feature contributions. To assess generalizability, we intend to validate our models on public HAR datasets like WISDM, PAMAP2, and UCI-HAR. Finally, we will explore real-time deployment and energy optimization to support practical use in mobile and wearable devices.

Supplemental Information

Supplemental Information 1 Feature Extraction.

Additional Information and Declarations

Competing Interests

Ivan Miguel Pires and Paulo Jorge Coelho are Academic Editors of PeerJ Computer Science.

Author Contributions

Luis Augusto Silva Zendron conceived and designed the experiments, performed the experiments, analyzed the data, performed the computation work, prepared figures and/or tables, authored or reviewed drafts of the article, and approved the final draft.

Paulo Jorge Coelho conceived and designed the experiments, performed the experiments, analyzed the data, performed the computation work, prepared figures and/or tables, authored or reviewed drafts of the article, and approved the final draft.

Christophe Soares conceived and designed the experiments, performed the experiments, analyzed the data, performed the computation work, prepared figures and/or tables, authored or reviewed drafts of the article, and approved the final draft.

Ivo Pereira conceived and designed the experiments, performed the experiments, analyzed the data, performed the computation work, prepared figures and/or tables, authored or reviewed drafts of the article, and approved the final draft.

Ivan Miguel Pires conceived and designed the experiments, performed the experiments, analyzed the data, performed the computation work, prepared figures and/or tables, authored or reviewed drafts of the article, and approved the final draft.

Ethics

The following information was supplied relating to ethical approvals (i.e., approving body and any reference numbers):

Ethics Committee from Universidade da Beira Interior approved the study with the reference CE-UBI-Pj-2020-035:ID1965.

Data Availability

The following information was supplied regarding data availability:

Data is available at Mendeley:

Pires, Ivan; Garcia, Nuno M. (2020), “Raw dataset with accelerometer, gyroscope and magnetometer data for activities with motion”, Mendeley Data, V2, doi: 10.17632/xknhpz5t96.2.

Code is available in the Supplemental Files.

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
