# Peer review of "Enhancing human activity recognition with machine learning: insights from smartphone accelerometer and magnetometer data"

_PeerJ Computer Science, doi:10.7717/peerj-cs.3137_

## Round 0.1 · original submission · Major Revisions

· Academic Editor

Major Revisions

Among the reviewers' comments, there are a couple of issues that must be carefully address: the related work (state-of-the-art) and the experimental sections.

Both are extremely important in a research paper and must be taken into account all the suggestions of the reviewers.

Reviewer 1 ·

Basic reporting

Overall, the paper is good, organised, well written and the content is significant for the field related to the problem-based learning approaches in computer science.
Some strengths related to both scientific content and form are:
- Based on previous similar works, the authors identified the unsolved challenges and proposed a study to examine how they could less influence the problem-based learning process.

Experimental design

yes

Validity of the findings

yes

Additional comments

na

·

Basic reporting

Enhancing human activity recognition with machine learning: Insights from smartphone accelerometer and magnetometer data

Human activity recognition (HAR) has evolved significantly due to advancements in sensor technology, AI, and machine learning (ML) algorithms. This paper aims to build on previous results by applying new techniques to the same dataset, enhancing outcomes from studies using Neural Networks, Random Forest, Support Vector Machine, CN2 rule inducer, Naive Bayes, and AdaBoost. The methodology includes smartphone sensor data collection, cleaning, normalization, feature extraction, and ML model implementations. The analysis indicated that Neural Network and Random Forest models performed well, with an AUC of 98.42%, classification accuracy of 90.14%, F1 score of 90.13%, precision of 90.18%, and recall of 90.14%.

The study looked technical and was well prepared. However, It has some significant concerns, as listed below:

1- The aim of the study, the gap in the literature, the main focus of the study, and the challenges of the previous works must be given in the introduction section. The novelty of the study is not explicit.

The authors stated that they aimed to use the previously obtained results to implement other techniques to analyze the same dataset and improve the results previously obtained in different studies.

Thus, the contributions of the study are not prominent.

The authors must explain the advantages of their model over the literature models.

3—The proposed model was not explained clearly. The explanations in the methods section are bookie information. The proposed model's details should be more detailed and technical.

4- The literature survey must extended using the most recent HAR studies, which use Deep learning.
The authors mentioned very few studies in the literature review section. It is far from taking a photo of literature.

In particular, 1D-CNN works that use raw signal data as input should be mentioned.

5—A general flow diagram figure must be added to the study.

6— The study must be checked grammatically.

7- Statistical significance analysis should be done.

8- The merits and demerits of the study should be given in the discussion section.

9- The results and discussion section must be separated into different subsections.

10- In the discussion section, the authors must give a table that depicts the results of the literature and the methods used.

Experimental design

I added all comments to the Basic Reporting section.

Validity of the findings

I added all comments to the Basic Reporting section.

Reviewer 3 ·

Basic reporting

Strengths:
The manuscript is well-structured and adheres to the PeerJ formatting guidelines.
The introduction provides a broad overview of Human Activity Recognition (HAR) and its applications, which is useful for contextualizing the research.
The literature review is fairly comprehensive and covers relevant previous works.
Weaknesses & Areas for Improvement:
Clarity and Conciseness: Some parts of the manuscript are verbose and can be rewritten more concisely to improve readability. Example:

Lines 36-50 (Introduction): The description of HAR applications is repetitive and could be condensed.
Lines 128-138 (Challenges & Considerations): The discussion on privacy concerns is useful, but it needs to be more structured to distinguish legal, technical, and ethical issues.
Suggested Improvement: Use bullet points or structured subsections to enhance clarity.

Figures and Tables:

Some figures (e.g., Figures 1-5) lack detailed captions explaining their significance.
The tables present well-organized results, but a summary table comparing different models based on computational efficiency, accuracy, and applicability in real-world scenarios would add value.
References & Citations:

While the manuscript includes relevant references, some recent deep learning approaches in HAR (e.g., transformer-based models, graph neural networks) are missing.
Citations such as Pires et al. (2018b, 2020a) are used multiple times. It is unclear whether these are the authors' previous works or a general reference in HAR research.
Suggested Improvement: Cite more recent studies on deep learning in HAR, especially CNNs, LSTMs, and transformer-based models.

Experimental design

Strengths:
The methodology follows a clear pipeline, covering data collection, preprocessing, model training, and evaluation.
The dataset used is publicly available, enhancing the reproducibility of the study.
The use of multiple classifiers (Neural Networks, Random Forest, Naïve Bayes, etc.) provides a comparative analysis of model performances.
Weaknesses & Areas for Improvement:
Choice of Machine Learning Framework:

The study uses Orange for model implementation, but it does not justify why this tool was chosen over alternatives like Scikit-learn, TensorFlow, or PyTorch.
Orange is GUI-based and does not allow much customization. This could be a limitation in real-world applications.
Suggested Improvement: Explain why Orange was used and discuss its advantages/disadvantages compared to Python-based ML libraries.

Feature Extraction:

The study extracts 15 features but does not provide a feature importance analysis. This makes it unclear which features contribute the most to model performance.
Peak detection algorithm is used, but how false peaks are handled is not described.
Suggested Improvement: Provide a feature importance ranking (e.g., using SHAP values or mutual information scores) to show which features are most significant.

Computational Performance:

The study does not discuss computational efficiency or model inference time.
Since HAR applications often require real-time processing, it would be useful to see a comparison of execution times for different classifiers.
Suggested Improvement: Add a table comparing training time, inference time, and memory usage for each model.

Validity of the findings

Strengths:
The results are clearly presented, with classification accuracy, AUC, precision, recall, and F1-score.
The Random Forest and Neural Network models show strong performance, which aligns with previous research in HAR.
Weaknesses & Areas for Improvement:
Overfitting & Generalization:

The high accuracy of some models raises concerns about possible overfitting.
No mention of regularization techniques used in Neural Networks.
It is unclear whether cross-validation folds were stratified.
Suggested Improvement: Include training vs. test accuracy to check for overfitting. Mention whether dropout, L2 regularization, or data augmentation were applied.

Error Analysis:

The paper does not provide a confusion matrix or detailed misclassification analysis.
Some activities (e.g., running vs. walking) may be confused, but this is not discussed.
Suggested Improvement: Add a confusion matrix to show which activities are misclassified. If available, provide examples of misclassified samples.

Comparison with State-of-the-Art:

The manuscript claims to improve previous results, but it lacks a comparison with state-of-the-art deep learning models.
No discussion on transfer learning or hybrid models combining sensor and video data.
Suggested Improvement: Compare results with recent transformer-based HAR models or CNN-based approaches.

Additional comments

Strengths:
The study contributes valuable insights into HAR using smartphone sensors and machine learning.
The dataset is well-documented and publicly available.
The comparative analysis of multiple classifiers is useful.
Weaknesses & Areas for Improvement:
Practical Applications & Real-World Impact:

The study should discuss how these HAR models could be deployed in real-world applications (e.g., healthcare, sports tracking, elderly monitoring).
No discussion on real-time implementation challenges.
Suggested Improvement: Discuss how the model can be deployed on mobile devices and whether battery consumption is a concern.

Ethical & Privacy Considerations:

HAR involves collecting sensitive data, but the study does not discuss privacy protection.
How does the study comply with GDPR or other data privacy laws?
Suggested Improvement: Briefly discuss privacy concerns and possible mitigation strategies (e.g., on-device processing, federated learning).

Reviewer 4 ·

Basic reporting

The authors need to conduct a comprehensive review of state-of-the-art Machine Learning (ML) and Deep Learning (DL) methods applied to the dataset used in this study. This review will help identify research gaps and compare the performance of different approaches, providing a solid foundation for further improvements.

I have not identified any significant novelty in this manuscript. The authors must clearly highlight the novelty of their proposed work, as the ML algorithms utilized in this study are widely adopted by numerous researchers in the field of Human Activity Recognition (HAR). Clearly defining the unique contributions will help distinguish this work from existing research.

Experimental design

It is recommended that the authors include a flow diagram to visually represent the proposed method. This would enhance clarity and provide a structured overview of the methodology, making it easier for readers to understand the workflow and key steps involved.

The authors have provided the configuration of the ML models in Table 1. However, it is unclear how these configurations were determined. Did the authors utilize any optimization technique to fine-tune these parameters? If so, they should provide a detailed explanation of the optimization process used. If no optimization technique was applied, the authors must clearly justify the criteria on which the configurations in Table 1 were selected.

Validity of the findings

The authors have validated the results using standard quality metrics. To enhance the interpretability of the findings, they are encouraged to implement suitable Explainable Artificial Intelligence (XAI) approaches. This would provide deeper insights into the model’s decision-making process and improve transparency.

Additional comments

The work has limited scope.

---

## Round 0.2 · Major Revisions

· Academic Editor

Major Revisions

The paper has been clearly improved during the last revision, nevertheless there are still some points that must be addressed in the new round. Please take into account carefully the suggestions of the reviewers

·

Basic reporting

In this version of the manuscript, the authors handled my issues. Thus, the study should be accepted.

Experimental design

In this version of the manuscript, the authors handled my issues. Thus, the study should be accepted.

Validity of the findings

In this version of the manuscript, the authors handled my issues. Thus, the study should be accepted.

Additional comments

In this version of the manuscript, the authors handled my issues. Thus, the study should be accepted.

Reviewer 3 ·

Basic reporting

The manuscript is generally well-written in professional and unambiguous English. The introduction effectively introduces the subject of smartphone-based human activity recognition (HAR), clearly articulating the motivation for the study. The background is thorough and supported with recent, relevant literature, especially on sensor fusion and mobile deployment constraints.

However, the following improvements are recommended:

Some sections are overly verbose and could benefit from condensation for improved readability (e.g., background paragraphs discussing sensor trends could be more concise).

The comparative literature table (Table 9) could be expanded to include more benchmark models for context.

Minor language polishing is needed in certain places for grammatical clarity (e.g., line 498: "The performance... is awful" could be revised to "The performance... is considerably lower").

Overall, the article meets the standard for basic reporting but would benefit from minor editorial improvements.

Experimental design

The experimental design is methodologically sound and falls within the scope of PeerJ Computer Science.

Strengths:

The dataset is publicly available and appropriately referenced.

Models are well described, with complete implementation details including hyperparameters and platform setup.

The study presents a reproducible visual workflow using Orange, which is commendable and supports transparency.

Areas for Improvement:

While the authors mention the use of Orange workflows, they should provide a link or DOI to the .ows workflow files used in the experiments.

External validation on a second dataset (e.g., WISDM or UCI-HAR) would strengthen the generalizability of findings.

Although the study mentions the exclusion of samples with missing values, a more detailed account of the preprocessing strategy and proportion of data affected would enhance reproducibility.

Validity of the findings

The findings are valid and well supported by experimental results.

Positive Aspects:

The use of lightweight and interpretable models like Random Forest and CN2 is appropriate for mobile deployment and explained clearly.

A thorough confusion matrix and statistical analysis (t-test, Wilcoxon, Friedman) reinforces the credibility of the reported results.

The inference time and model footprint analysis adds real-world value and supports the feasibility of mobile application.

Suggested Enhancements:

While the study claims reproducibility, it lacks implementation of model-agnostic XAI tools (e.g., SHAP, LIME), which the authors acknowledge. Including even a basic SHAP plot would substantiate the claim of explainability.

The limitations section (6.7) should more explicitly outline constraints such as device heterogeneity, energy consumption variability, and noise conditions in uncontrolled environments.

Additional comments

The authors have presented a novel and useful contribution by demonstrating high classification performance using non-normalized smartphone sensor data and lightweight machine learning models.

The practical emphasis on deployment efficiency is timely and relevant.

I commend the authors for addressing both performance and interpretability, which are often overlooked in similar HAR studies.

·

Basic reporting

The work, "Enhancing Human Activity Recognition with Machine Learning: Insights from Smartphone Accelerometer and Magnetometer Data” is interesting because it contributes novel insights and a wide range of experimentation.

The inclusion of sections on computational cost, replicability, and comparison with state-of-the-art research is essential for studies of this nature, and I thank the authors for including them.

There are some considerations that I think need to be included and polished in the manuscript.

1. In the introduction, the bullet points outlining the main contributions of this research contain two sentences that require a reference to justify the assertion with a reference or should be deleted. "*Uncommon in HAR" and "*Most studies use accelerometer + gyroscope or omit magnetometer."
2. When the Regulation GRPT is mentioned in section 2.3. It would be appropriate to mention the area in which the regulation applies, i.e., the European Union.

Experimental design

The experiments are well-designed and executed, and the results are interesting. However, it is recommended that the software version and the SciPy library be included in Section 3.3.1 because they are used for data preprocessing.

Validity of the findings

The comparition between the models in the experimentations and between this work and the related works are conduced only by observation, without including any statistical test. I understand that may be difficult to find the raw results of the related works for perform a statistical test, but it a must include when it is possible. I suggest the authors to made statistical tests between the models used and publish the raw results for the future works whose authors want to compare their results with the yours.

---

## Round 0.3 · Minor Revisions

· Academic Editor

Minor Revisions

Please read carefully the review comments of the reviewers. It is paramount for reproducibility the availability of the datasets.

Reviewer 3 ·

Basic reporting

The manuscript is written in clear, professional English. The language has improved from the previous version, with minor revisions still needed to enhance clarity and conciseness. The Introduction and Background sections adequately contextualize the study and include sufficient and relevant references to recent literature. The structure adheres to PeerJ standards and disciplinary norms.

Literature is up to date and appropriately cited.

The figures and tables are relevant and well-labeled.

Data availability is addressed, and a Zenodo link for Orange workflows was added as per reviewer recommendation.

Experimental design

The experimental design is solid and replicable. The authors use a publicly available dataset, perform thorough preprocessing, and implement various traditional ML models. They correctly justify the choice of lightweight models over deep learning methods for mobile deployment. The introduction of non-normalized data and use of magnetometer signals adds novelty to the study.

Methodology is sufficiently detailed.

Model hyperparameters and implementation platforms are included.

The experimental protocol is transparent and reproducible.

Orange visual workflows support open science practices.

Validity of the findings

The findings are valid, well-supported by evaluation metrics (accuracy, F1, AUC, etc.), and demonstrated using a reproducible pipeline. The paper’s core strength lies in the demonstration that classical ML models can rival deep learning for HAR tasks under real-world mobile constraints.

Results align with the objectives stated in the Introduction.

Statistical testing supports validity of comparisons.

Model performance is benchmarked and summarized clearly.

Additional comments

This manuscript offers a valuable contribution to the Human Activity Recognition (HAR) field by presenting a robust, lightweight, and efficient alternative to deep learning models. The focus on magnetometer integration and deployment efficiency is timely and highly relevant for mobile health, fitness, and security applications.

Strengths:

Use of non-normalized data for resource efficiency

Incorporation of magnetometer data (often underutilized)

Reproducible implementation using Orange workflows

Thorough comparison with state-of-the-art HAR approaches

Strong statistical validation of model results

Expanded limitations section and open benchmarking initiative

Areas for Improvement:

Consider including basic XAI support (e.g., SHAP summary plot) to substantiate interpretability claims.

Provide a more detailed breakdown of data cleaning steps and proportion of excluded samples in preprocessing.

Ensure all language is formal and appropriate for an academic audience (remove “awful” and similar phrasing).

In future work, external validation on public benchmark datasets (e.g., WISDM) is encouraged.

·

Basic reporting

The paper has improved in the required areas. The improvement in the work is appreciated.

Experimental design

No comments

Validity of the findings

No comments

Additional comments

The data reference points to a repository that has no data, only a README. This would need to be fixed before publishing the paper.

---

## Round 0.4 · accepted · Accept

· Academic Editor

Accept

All the issues highlighted by the reviewers have been addressed.

·

Basic reporting

I appreciate the Zenodo repository being fixed. But the mendeley repository returns "The dataset you are looking for was not found." Check the visibility.

Experimental design

Nothing to comment on. Good work.

Validity of the findings

Nothing to comment on. Good work.

Additional comments

Nothing to comment on. Good work.